# Comparative Transcriptome and Widely Targeted Metabolome Analysis Reveals the Molecular Mechanism of Powdery Mildew Resistance in Tomato

**DOI:** 10.3390/ijms24098236

**Published:** 2023-05-04

**Authors:** Wenjuan Liu, Xiaomin Wang, Lina Song, Wenkong Yao, Meng Guo, Guoxin Cheng, Jia Guo, Shengyi Bai, Yanming Gao, Jianshe Li, Zhensheng Kang

**Affiliations:** 1College of Enology and Horticulture, Ningxia University, Yinchuan 750021, China; 2Ningxia Modern Facility Horticulture Engineering Technology Research Center, Yinchuan 750021, China; 3Key Laboratory of Modern Molecular Breeding for Dominant and Special Crops in Ningxia, Yinchuan 750021, China; 4State Key Laboratory of Crop Stress Biology for Arid Areas, College of Plant Protection, Northwest A&F University, Yangling 712100, China

**Keywords:** tomato, powdery mildew, transcriptome, widely targeted metabolome, combined analysis

## Abstract

Powdery mildew is a serious problem in tomato production; therefore, the PM-resistant tomato inbred line, ‘63187’, and the susceptible tomato variety, ‘Moneymaker (MM)’, were used as experimental materials for the combined analysis of transcriptome and widely targeted metabolome on tomato leaves at 0 h post inoculation (hpi), 12 hpi, and 48 hpi. The results indicated that 276 genes were expressed in all treatments, and the K-means cluster analysis showed that these genes were divided into eight classes in ‘63187’ and ten classes in ‘MM’. KEGG enrichment showed that amino acid metabolism, signal transduction, energy metabolism, and other secondary metabolites biosynthesis pathways were significantly enriched. Interestingly, the analysis of WRKY family transcription factors (TFs) showed that the expression of four TFs in ‘63187’ increased with no obvious change in ‘MM’; and the expression of one TF in ‘MM’ increased with no obvious change in ‘63187’. The combined analysis revealed that both phenylpropanoid biosynthesis and flavonoid biosynthesis pathways were enriched in ‘63187’ and ‘MM’. In ‘63187’, six metabolites involved in this pathway were downregulated, and four genes were highly expressed, while in ‘MM’, three metabolites were upregulated, four metabolites were downregulated, and ten genes were highly expressed. These metabolites and genes might be candidates for PM resistance or susceptibility in subsequent studies. These results provide favorable molecular information for the study of the different resistances of tomatoes to PM, and they provide a basis for the breeding of tomato varieties resistant to PM.

## 1. Introduction

Tomato powdery mildew (PM) is a widely distributed and rapidly spreading fungal disease. It first broke out in Taiwan Province, China, in 1919. It is caused by *Erysiphe Polygoni* DC, *Leveillula Taurica*, *Oidium lycopersici* (*Ol*), or *Oidium neolycopersici* (*On*), with *On* as the main pathogen [1]. During the early stage of *O. neolycopersici* infection, tomato leaves have faded green spots. When they turn yellow, white spore spots gradually appear; then, they enlarge and ultimately cover the entire leaf. In the late stage of the disease, tomato leaves become yellow and curl, weakening their photosynthetic capacity and seriously affecting tomato production in greenhouses [2]. The use of chemicals in agricultural production is the most effective measure to control PM. Although this measure can inhibit the disease in a short period of time, it also leads to drug resistance to pathogens. PM control could also be problematic in the organic production of tomatoes, as most of the biologicals may not be effective. Therefore, in recent years, with the constantly development of molecular biology technology and the deepening of molecular breeding work, screening disease-resistant genes and breeding disease-resistant varieties have become more effective, environmentally friendly, and economical approaches to prevent PM [3].

Although most cultivated tomato varieties are highly susceptible to disease, researchers have found resistance resources from wild tomatoes [4]. Thus far, a total of five genes of dominant resistance (*Ol-1*, *Ol-3*, *Ol-4*, *Ol-5*, and *Ol-6*) have been obtained from wild tomato resources; one recessive resistance gene (*Ol-2*) and three resistance quantitative traits locus (QTLs) were identified [5,6]. In the past, forward and reverse genetics were used to identify candidate genes for resistance to PM [7,8,9]. RNA sequence (RNA-seq), a novel technique for transcriptome analysis, has been extensively adopted to recognize main pathways and response genes under biological or abiotic stresses, accurately measure transcription levels, and reveal response mechanisms to particular stimuli [10]. At present, this technique has been used to study PM resistance genes and regulation mechanisms in horticultural crops such as cucumber, melon, and rose [11,12,13]. Metabolites are the foundation of biological phenotype, through which biological processes and mechanisms can be intuitively and effectively understood. In recent years, transcriptome and metabolome data have been used to analyze the internal changes of organisms at both cause and effect levels, identify important genes, metabolites, and metabolic pathways, construct core regulatory networks, and reveal the responses of plants to biological stresses and the interactions between plants and pathogens [14]. For instance, Yang et al. [15] found that the biosynthesis of flavonoids played an essential part in the resistance of *camellia oleifera* to anthracnose by analyzing the co-expressed genes and co-accumulated metabolites between resistant and susceptible materials. Zhao et al. [16] analyzed the pathway of 2560 co-expressed genes in two varieties—resistant- and susceptible-to-wheat-stripe-rust—and found that most genes were closely related to defense and metabolic activities. Yuan et al. [17] identified 3418 genes and 405 metabolites between two varieties of resistant- and susceptible-to-PM *Tibetan hulless barley*. They found a corresponding relationship between gene expression and metabolic spectrum, and the activated defenses lead to changes in the metabolites involved in the plant defense response. Therefore, integrated transcriptome and metabolome analysis can offer a greater insight into plant responses to pathogens; search for candidate resistance-related genes, metabolites, and metabolic pathways, which is conducive to the prevention and control of diseases; and increase crop yield and farmers’ economic benefits. However, there was relatively little research on the mechanism of PM resistance in tomatoes. The candidate genes, metabolites, and metabolic pathways related to the resistance and susceptibility of tomato to PM, screened by combined analysis of transcriptome and metabolome, have not been reported yet.

Therefore, in this study, the PM-resistant tomato inbred line, ‘63187’, and the susceptible tomato variety, ‘Moneymaker (MM)’, were used as experimental materials to carry out the combined analysis of transcriptome and widely targeted metabolome on tomato leaves at 0 h post inoculation (hpi), 12 hpi, and 48 hpi. It was expected that by digging differentially expressed genes (DEGs) and differentially accumulated metabolites (DAMs), the key candidate genes, metabolites, and metabolic pathways of tomato to PM could be identified, and regulatory metabolic pathways and the molecular mechanisms of tomato resistance and susceptibility to PM could be systematically and comprehensively analyzed. The results would provide favorable molecular information for the response of different resistant tomato leaves’ inoculation with *O. neolycopersici* and a basis for breeding PM-resistant tomato varieties.

## 2. Results

### 2.1. Phenotype Changes of Two Tomato Materials after Inoculation with O. neolycopersici

After 25 days of inoculation with *O. neolycopersici* on tomato leaves, the incidence rate was counted, and the disease index of the two materials was calculated according to the disease grading standard. It was found that ‘MM’ and ‘63187’ displayed diverse resistance levels to PM. ‘63187’ was more resistant to PM, and its growth was better than ‘MM’. The white spore spots on the leaf surface of ‘MM’ almost covered the whole leaf of all leaves, while ‘63187’ only had sporadic white spore spots. The incidence rates of ‘MM’ and ‘63187’ were 73.60% and 23.64%, and the disease indices were 66.72% and 19.83%, respectively. According to the classification criteria of group disease resistance, ‘MM’ was high-sensitivity (HS) and ‘63187’ was resistant (R) (Figure 1).

### 2.2. Transcriptome Analysis

For screening candidate genes associated with PM resistance, exploring the molecular mechanisms of different resistant tomatoes under *O. neolycopersici* stress, and gaining a holistic overview of the gene expression dynamics of tomato leaves after inoculation with *O. neolycopersici*, the cDNA libraries of leaves of two tomato varieties above inoculated with *O. neolycopersici* at 0 hpi (CK), 12 hpi, and 48 hpi were sequenced by transcriptome. The samples of PM-resistant tomato inbred line ‘63187’ were represented by R0, R12, and R48, respectively, and the samples of susceptible tomato variety ‘MM’ were represented by S0, S12, and S48, respectively. A total of 47, 875, 617 raw reads were obtained by transcriptome sequencing, and 46, 635, 723 clean reads were derived after filtering the original data; altogether, 125.9 Gb of clean data was acquired. Each sample had over 6 Gb of clean data and the GC content was about 42.87%. The percentage of Q20 base was about 97.24%, and the Q30 base was about 91.91%, suggesting the relatively high quality of transcriptome data, which could be utilized for further data analysis (Appendix A).

Clean reads were mapped to the reference genome, which was 96.99% aligned with the tomato reference genome; among them, the only reads mapped to the reference genome accounted for 95.26% of the clean reads, and the multiple reads mapped to the reference genome accounted for 1.73% of the clean reads (Appendix A). Among them, 90.97% were in located in exons, 3.93% in introns and 4.99% in intergenic regions. R^2^ > 0.87 in this study indicated good repeatability and reliability of transcriptome samples (Appendix A). As shown in Figure 2A, the two materials showed obvious separation at different time points after inoculation. There were similar PC values between the three biological replicates at the same time point of each material, indicating that the sample repeatability was good and consistent with the heat map results (Figure 2B).

#### 2.2.1. Identification and Annotation of DEGs

This study screened a total of 9855 DEGs. Taking R0 and S0 as CK, 6125 DEGs were found in ‘63187’ and 7786 in ‘MM’. In ‘63187’, 2560 and 872 DEGs were upregulated, and 2217 and 476 DEGs were downregulated at 12 hpi and 48 hpi, respectively. In ‘MM’, 3025 and 1540 DEGs were upregulated, and 2487 and 734 DEGs were downregulated at 12 hpi and 48 hpi, respectively (Figure 2C). In both ‘63187’ and ‘MM’, DEGs number at 12 hpi was higher than that at 48 hpi. A Venn diagram showed that 790, 266, 1364, and 739 DEGs screened were specifically expressed in four comparison groups, R0_vs_R12, R0_vs_R48, S0_vs_S12, and S0_vs_S48, respectively. In addition, 276 DEGs were co-expressed in all treatments, and 599 and 1023 DEGs were co-expressed in ‘63187’ and ‘MM’, respectively (Figure 2D).

A total of 994 DEGs were screened in S12_vs_R12, of which 609 DEGs were upregulated and 385 were downregulated; 1114 DEGs were screened in S48_vs_R48, of which 676 were upregulated and 438 were downregulated. A total of 5469 DEGs were screened in R12_vs_R48, of which 2871 were upregulated and 2598 were downregulated; 6102 DEGs were screened in S12_vs_S48, of which 3284 were upregulated and 2817 were downregulated (Appendix A). The Venn diagram showed a total of 4368 DEGs in R12_vs_R48 and S12_vs_S48 (Appendix A) and a total of 564 DEGs in S12_vs_R12 and S48_vs_R48 (Appendix A).

#### 2.2.2. Functional Annotation and Classification of DEGs

In order to further understand the role of DEGs, 276 DEGs from four comparison groups were annotated and classified through Gene Ontology (GO) (Appendix A). Among them, cellular anatomical entity (GO:0110165) was the most abundant term in cell composition, cellular processes (GO:0009987) and metabolic processes (GO:0008152) were the most prominent terms in biological processes, and binding (GO:0005488) and catalytic activity (GO:0003824) were the highest-expressed terms in molecular function.

In order to further clarify the roles of DEGs in PM, an enrichment analysis of 276 DEGs from four comparison groups was performed using the Kyoto Encyclopedia of Genes and Genomes (KEGG) database (Figure 3A). Among them, plant–pathogen interaction (ko04626), phenylpropane biosynthesis (ko00940), MAPK signaling pathway–plant (ko04016), and flavonoid biosynthesis (ko00941) were all enriched on the KEGG pathway. In addition, notably enriched pathways were photosynthetic antenna proteins (ko00196), biosynthesis of amino acids (ko01230), metabolic pathways (ko01100), 2-oxocarboxylic acid metabolism (ko01210), glucosinolate biosynthesis (ko00966), valine, leucine, and isoleucine biosynthesis (ko00290), tropane, piperidine and pyridine alkaloid biosynthesis (ko00960), and plant hormone signal transduction (ko04075).

In particular, we discovered that the genes participated in the synthesis of other secondary metabolites pathways in the four comparison groups of tomato leaves after inoculation with *O. neolycopersici* were *cevi16*, *HCT*, *PER12*, *AnthOMT*, *TR1*, and *TRN1*. The genes involved in signal transduction included *AX15A*, *BAK1*, *IAA19*, *SAUR21*, *PIF3*, *ARR-B*, *JAZ*, *SAUR76*, *TGA1*, *BRI1*, *MYC2*, *SAUR36*, *SAUR24*, *Prg1*, *SAUR67*, *WRKY35*, *ANP1*, *CHIB*, *FLS2*, and *FLS2X2*. The genes involved in environmental adaptation include *PTI1*, *CML*, *CNGC1*, *PIK1*, and *WRKY2*. The genes involved in amino acid metabolism were *DAHPS*, *AS*, *ASNS*, *ARGE*, *PK*, *TD2*, *BCAT2*, and *UGT74B1*. The genes involved in energy metabolism included *CAB1B*, *CAB1C*, *CAB3C*, *CAB6A*, and *CAB12* (Appendix A).

#### 2.2.3. K-Means Clustering Analysis of DEGs

In order to study the gene expression patterns under different treatment conditions, K-means clustering analysis was conducted on 276 DEGs. It was found that 276 DEGs were divided into 8 classes in ‘63187’ and 10 classes in ‘MM’. In ‘63187’, the expression of 60 DEGs of subclass 6 increased first and then decreased after inoculation with *O. neolycopersici*, 38 DEGs of subclass 7 were continuously decreased, and 10 DEGs of subclass 8 were continuously increased, while the rest of the subclasses exhibited the trend of increased and then decreased, but the amplitudes of increase and decrease were different (Appendix A). In ‘MM’, 7 DEGs of subclass 3 and 10 DEGs of subclass 8 were continuously increased after inoculation with *O. neolycopersici*, while the 38 DEGs of subclass 7 were continuously decreased, and the 25 DEGs of subclass 6 and 34 DEGs of subclass 10 were all firstly decreased and then increased. Similarly, the other subclasses all increased and then decreased (Appendix A). In addition, we also found that in ‘MM’, among the ten DEGs of subclass 1, the expression of five DEGs in ‘MM’ was continuously increased, but they first increased and then decreased in ‘63187’. The continuously increased and decreased of subclasses in ‘63187’ and ‘MM’ were taken as a Venn diagram (Appendix A); we found that seven specific DEGs were continuously increased in ‘63187’, and thirteen specific DEGs were continuously increased in ‘MM’ after inoculation with *O. neolycopersici*. Only 1 DEG of 276 DEGs was continuously decreased in ‘63187’ but continuously increased in ‘MM’. The expression of this gene at 12 hpi was close to 0 hpi expression in ‘63187’, while 5.03-fold that of 0 hpi in ‘MM’; at 48 hpi, its expression was 0.36-fold that of 0 hpi in ‘63187’ and 7.02-fold that of 0 hpi in ‘MM’ (Appendix A).

#### 2.2.4. Transcription Factor Analysis of DEGs

A total of 796 transcription factors (TFs) were annotated in 9855 DEGs, and these TFs were divided into 80 transcription families. The first three families with the maximum number of TFs were the bHLH family, AP2/ERF-ERF family, and WRKY family, which contained 49, 48, and 40 transcription factors, respectively (Appendix A). WRKY is one of the biggest plant-specific TFs families and has a key function in plant growth, development, and stress response, with particular reference to disease resistance [18]. Therefore, we further analyzed the WRKY family genes. The findings revealed that 4 TFs were highly expressed in S0, 14 in S12, 2 in R0 and R12, respectively, 8 in R48, and 10 both in S12 and R12 (Figure 3B). Further analysis of the results revealed that after inoculation with *O. neolycopersici*, the expression levels of *WRKY3* (*Solyc09g015770.3*), *WRKY46* (*Solyc08g067340.3*), *WRKY51* (*Solyc04g051690.3*), and *WRKY70* (*Solyc03g095770.3*) continuously increased only in ‘63187’; at 48 hpi, the expression levels of these genes were 3.18-fold, 2.78-fold, 3.29-fold, and 3.27-fold those of 0 hpi, respectively. However, the expression of these genes did not change dramatically in ‘MM’ after inoculation with *O. neolycopersici*. The expression level of *WRKY13* was continuously increased in ‘MM’ after inoculation, the expression levels of this gene at 12 hpi and 48 hpi were 3.21-fold and 3.29-fold those of 0 hpi in ‘MM’; however, the performance of this gene did not change in any significant way in ‘63187’ (Appendix A).

#### 2.2.5. Verifying DEGs by qRT-PCR

To verify the reliability of the data expression level of RNA-seq, the qRT-PCR method was used to evaluate nine candidate genes (Figure 3C). The obtained qRT-PCR and transcriptome data were analyzed for linear correlation. The results indicated that the correlation coefficient between the qRT-PCR and transcriptome data was R^2^ = 0.7647; for ‘63187’, the correlation coefficient between the two was R^2^ = 0.4914; and for ‘MM’, the correlation coefficient between the two was R^2^ = 0.5886. The relevance was relatively good (Appendix A). The expression trend of the above qRT-PCR results was highly consistent with RNA seq, indicating the accuracy of transcriptome sequencing results.

From Figure 3C, it can be seen that after inoculation with *O. neolycopersici*, the expression levels of *R2R3MYB* (*Solyc03g093890.3*), *SGT1* (*Solyc12g057070.2*), *SAR2* (*Solyc01g06130.3*), *WRKY13* (*Solyc04g051540.3*), *PLAT1* (*Solyc03g096550.3*), and *HMGR* (*Solyc02g082260.3*) in ‘MM’ and ‘63187’ increased compared to 0 hpi; the expression level of *AnthOMT* (*Solyc09g082260.3*) in ‘MM’ increased compared to 0 hpi, while the expression level in ‘63187’ decreased compared to 0 hpi; the expression level of *WRKY46* (*Solyc08g067340.3*) in ‘MM’ decreased compared to 0 hpi, while the expression level in ‘63187’ increased compared to 0 hpi; the expression level of *WRKY30* (*Solyc10g009550.3*) increased in ‘MM’ compared to 0 hpi, decreased in ‘63187’ at 12 hpi compared to 0 hpi, and increased at 48 hpi compared to 0 hpi. The expression of ‘63187’ was reduced by 0.05 and 0.37-fold compared to ‘MM’ after 12 hpi in *AnthOMT* and *WRKY13*, and by 0.02 and 0.40-fold compared to ‘MM’ after 48 hpi. The expression to ‘MM’ ratio of ‘63187’ was almost unchanged 12 hpi of *WRKY46*, *R2R3MYB,* and *SAR2*, and increased 2.53, 2.01, and 3.19-fold after 48 hpi. After 12 hpi of *WRKY30* and *SGT1*, the expression of ‘63187’ was reduced by 0.35 and 0.20-fold compared to ‘MM’, and after 48 hpi, the expression of ‘63187’ was almost unchanged compared to ‘MM’. The expression of ‘63187’ was almost unchanged from ‘MM’ ratio after 12 hpi and 48 hpi of *PLAT1*. After 12 hpi of *HMGR*, the expression to ‘MM’ ratio of ‘63187’ was almost unchanged and after 48 hpi, the expression to ‘MM’ ratio of ‘63187’ was reduced by 0.21-fold (Appendix A).

### 2.3. Metabolome Analysis

In order to screen candidate metabolites related to PM, tomato leaves from ‘MM’ and ‘63187’ were sampled at 0 hpi, 12 hpi, and 48 hpi. Then, the widely targeted metabolite sequencing was performed. On the basis of the UPLC-MS/MS testing platform and the MWDB component database, a total of 1058 metabolites were measured, which were grouped into 10 sections, including flavonoids (18.05%), phenolic acids (16.54%), alkaloids (13.89%), lipids (14.08%), others (10.11%), amino acids and their derivatives (8.13%), organic acids (7.09%), nucleotides and their derivatives (6.24%), lignans and coumarin (4.54%), and terpenoids (1.32%) (Figure 4A).

To ensure the repeatability of the analysis process, a quality control sample (QC) was inserted for each of the 10 samples taken during the instrument analysis. Both metabolite extraction and assay repeatability can be determined by analyzing the total ion current (TIC) overlap of the different QC centers. The data revealed significant overlap in the curves of the TIC assay for metabolites; the holding time and intensity of the peak were uniform, indicating that the mass spectrometer had better signal stability when detecting the same sample at various times (Appendix A). The high instrument reliability offered an essential guarantee of reproducible and reliable data. The Pearson correlation coefficient was determined to reflect the repeatability of the sample within the group (Figure 4B and Appendix A). The principal component analysis (PCA) results showed excellent repeatability of samples within groups and a separation between the groups (Figure 4C). In supervised Orthogonal Partial Least Squares Discriminant Analysis (OPLS-DA), the values of R^2^Y and Q^2^ in R0_vs_R12, R0_vs_R48, S0_vs_S12 and S0_vs_S48 were 1.000 and 0.898, 0.999 and 0.928, 0.999, and 0.893, 1.000, and 0.899, respectively (Appendix A). The results suggested that the applied model was highly reliable, and the data were available for the next step of the analysis.

#### 2.3.1. Identification of Differentially Accumulated Metabolites

Taking R0 and S0 as controls, 451 and 468 DAMs were found in ‘63187’ and ‘MM’, respectively. In ‘63187’, 110 DAMs were upregulated, and 54 DAMs were downregulated at 12 hpi, and 181 were upregulated and 106 DAMs were downregulated at 48 hpi, respectively. In ‘MM’, 96 DAMs were upregulated, and 96 DAMs were downregulated at 12 hpi, and 149 were upregulated and 127 DAMs were downregulated at 48 hpi, respectively (Appendix A). A Venn diagram showed that 27, 84, 52, and 76 DAMs screened out were especially accumulated in R0_vs_R12, R0_vs_R48, S0_vs_S12, and S0_vs_S48, respectively. In addition, 46 DAMs were co-accumulated in all treatments, and 95 and 87 DAMs were co-accumulated in ‘63187’ and ‘MM’, respectively (Figure 4D).

A total of 172 DAMs were screened in S12_vs_R12, of which 80 were upregulated and 92 were downregulated; 173 DAMs were screened in S48_vs_R48, of which 70 were upregulated and 103 were downregulated; 244 DAMs were screened in R12_vs_R48, of which 145 were upregulated and 99 were downregulated; 248 DAMs were screened in S12_vs_S48, of which 131 were upregulated and 117 were downregulated (Appendix A). The Venn plot showed a total of 142 DAMs in R12_vs_R48 and S12_vs_S48 (Appendix A), and a total of 90 DAMs in S12_vs_R12 and S48_vs_R48 (Appendix A).

In this study, the top 10 metabolites with the largest absolute value of log_2_FC were screened in R0_vs_R12. Among them, one downregulated phenolic acid was 2-phenylethanol and two upregulated were furanfructose-α-D-(6-mustard acyl) and drocaffeoylglucose. The three upregulated terpenoids were 2α-hydroxyursolic acid, alphitolic acid, and 3,24-dihydroxy-17,21-semiacetal-12(13)oleanolic fruit. Two downregulated lignans and coumarins were dihydrodehydrodiconiferyl alcohol-4-O-glucosidedihydrodehy and isolariciresinol-9′-O-glucoside. One upregulated flavonoid substance was alfalfa-7-O-(2″-feruloyl)glucoside-5-O-glucoside, and one downregulated lipid was lysoPC 18:1 (Appendix A, Appendix A). In R0_vs_R48, the top 10 metabolites with the largest absolute value of log_2_FC were screened, among which three upregulated terpenoids were 2α-hydroxyursolic acid, 3,24-dihydroxy-17,21-semiacetal-12(13)oleanolic, and alphitolic acid. Two downregulated organic acids were glutaric acid* and 2-Oxoadipic acid. Two other substances upregulated were D-galacturonic acid and D-glucuronic acid. The upregulated phenolic acids and flavonoids were dihydrocaffeoyl glucose and tricin-7-O-(2″-feruloyl) glucoside-5-O-glucoside, respectively; the downregulated lipid was 1-Octadecanol (Appendix A, Appendix A).

In S0_vs_S12, the top 10 metabolites with the largest absolute value of log_2_FC were screened and five upregulated terpenoids were 3,24-dihydroxy-17,21-semiacetal-12(13)oleanolic fruit, 2α,3α,23-trihydroxyolean-12-en-28-oic acid, 2α-Hydroxyursolic acid, alphitolic acid, and madasiatic acid. The upregulated nucleotides and derivatives, lipids, and alkaloids were xanthine, 9-Oxo-12Z-octadecenoic acid, and α-solasonine, respectively. One upregulated flavonoid was luteolin-7-O-rutinoside-5-O-rhamnoside and one downregulated was kaempferol-3-O-(2″-O-acetyl)glucuronide (Appendix A, Appendix A). In S0_vs_S48, the top 10 metabolites with the largest absolute value of log_2_FC were screened; five upregulated terpenoids were alphitolic acid, 3,24-Dihydroxy-17,21-semiacetal-12(13)oleanolic fruit, 2α,3α,23-trihydroxyolean-12-en-28-oic acid, 2α-hydroxyursolic acid, and madasiatic acid. The downregulated organic acids and other substances were citraconic acid and butyl beta-D-glucoside, respectively. The upregulated alkaloids, nucleotides and their derivatives, and flavonoids were α-solasonine, cytidine 5’-monophosphate (cytidylic acid), and luteolin-7-O-rutinoside-5-O-rhamnoside, respectively (Appendix A, Appendix A). In conclusion, the accumulation of terpenoids 2α-hydroxyursolic acid, alphitolic acid, 3,24-Dihydroxy-17,21-semiacetal-12(13)oleanolic fruit, and flavonoids luteolin-7-O-rutinoside-5-O-rhamnoside may have a positive role in tomato leaves response to PM.

#### 2.3.2. K-Means Cluster Analysis of DAMs

Through K-means clustering analysis, 46 common DAMs were classified into 9 categories, both in ‘63187’ and ‘MM’ (Appendix A). Alphitolic acid, 2α-Hydroxyursolic acid, 9-Hydroxy-12-oxo-15(Z)-octadecenoic acid, tricin-4′-O-glucoside-7-O-glucoside, N-Acetyl-L-threoninein, 3,24-Dihydroxy-17,21-semiacetal-12(13)oleanolic fruit, and (22R,25R)-16β-H-22a-N-Spirosol-3β-ol-5-ene-3-O-rhamnosyl(1→2)[rhamnosyl(1→4)]glucoside continued to increase in ‘63187’, and first increased and then decreased in ‘MM’. The lysoPE 16:3 was continuously decreased in ‘MM’, while it was decreased at first and then increased in ‘63187’. The above eight metabolites included three terpenoids, two lipids, one alkaloid, one flavonoid, and one amino acid and its derivatives. Isorhamnetin-3-O-(6’’-O-p-coumarol) sophoroside-7-O-rhamnoside, luteolin-7-O-rutinoside-5-O-rhamnoside, and shikimic acid were continuously increased in ‘MM’, but they were increased firstly and then decreased in ‘63187’. The above three metabolites included two flavonoids metabolites and one organic acid (Appendix A).

### 2.4. Correlation Analysis of DEGs and DAMs

For the purposes of determining the association between DEGs and DAMs of different resistant tomato materials inoculated with *O. neolycopersici*, compensating for data problems due to data loss, noise, and other factors in single-omics analysis, and reducing the occurrence of false positives, a combined analysis of transcriptome and metabolome data was carried out, which was conducive to the phenotype study of biological models and their relationship with the adjustment mechanisms of biological processes. The Pearson correlation algorithm was used to compute the relationship between candidate metabolites and genes, and a network diagram of candidate metabolites and genes was drawn (Appendix A, Appendix A). The results showed that (22R,25R)-16β-H-22a-N-Spirosol-3β-ol-5-ene-3-O-rhamnosyl(1→2)[rhamnosyl(1→4)]glucoside and 2α-Hydroxyursolic acid were positively correlated with *SGT1*, *Solyc11g022590.1*, *SLC15A3_4*, *PHT*, *CTSH*, *RMA1H1*, *ARPI*, *PLAT1*, *Solyc09g073030.3*, *DAD1*, *HMGR*, *Solyc03g123390.3*, *RNASET2*, *Solyc02g063390.3*, and *IBTK*. Shikimic acid and luteolin-7-O-rutinoside-5-O-rhamnoside were positively correlated with *RMA1H1*, *HMGR*, *Solyc03g123390.3*, *SLC15A3_4*, *PHT*, *SGT1*, *Solyc11g022590.1*, *CTSH*, *Solyc02g063390.3*, *ARPI*, *RNASET2*, *Solyc09g073030.3*, *Solyc05g054320.3*, and *PLAT1*. Alphitolic acid and 3,24-Dihydroxy-17,21-semiacetal-12(13) oleanolic fruit were positively correlated with *SGT1*, *PLAT1*, *Solyc09g073030.3*, *Solyc11g022590.1*, *RMA1H1*, and *SLC15A3_4*, *PHT*. 9-Hydroxy-12-oxo-15(Z)-octadecenoic acid was positively correlated with *Solyc09g073030.3* and *Solyc02g063390.3*. N-Acetyl-L-threonine was positively correlated with *Solyc09g073030.3* and *IBTK*. Isorhamnetin-3-O-(6″-O-p-coumarol) sophoroside-7-O-rhamnoside was positively correlated with *Solyc03g123390.3* and *HMGR*. Tricin-4′-O-glucoside-7-O-glucoside was positively correlated with *SGT1*, *Solyc11g022590.1*, *Solyc09g073030.3*, *RMA1H1*, *SLC15A3_4*, *PHT*, and *RNASET2*.

*SGT1*, *Solyc11g022590.1*, *SLC15A3_4*, *PHT*, *CTSH*, *RMA1H1*, *HMGR*, *ARPI*, and *Solyc03g123390.3* were actively correlated with 9-Hydroxy-12-oxo-15(Z)-octadecenoic acid and N-Acetyl-L-threonine; they were negatively correlated with lysoPE 16:3. *Solyc05g054320.3* was positively correlated with 2α-Hydroxyursolic acid and negatively correlated with lysoPE 16:3. *DTX29* was actively correlated with shikimic acid and negatively correlated with lysoPE 16:3. *TPS37* was positively correlated with 2α-Hydroxyursolic acid and shikimic acid. *DAD1* was positively correlated with shikimic acid. *RNASET2* was negatively correlated with lysoPE 16:3.

#### 2.4.1. KEGG Enrichment Analysis of Co-Expressed DEGs and DAMs

The results of the extensive target metabolome analysis were combined with transcriptome analysis to map both DEGs and DAMs in the same comparison group to the KEGG pathway to better understand the relationship between genes and metabolites. Interestingly, the results demonstrated that phenylpropane biosynthesis (ko00940) and flavonoid biosynthesis (ko00941) pathways were both abundant in ‘63187’ and ‘MM’. A network of the diagram was constructed to further analyze the relationship between DAMs and DEGs in the phenylpropane biosynthesis and flavonoid biosynthesis pathways by combining all the metabolites and genes result (Figure 5A). The results showed that *HCT* (*Solyc06g051320.3* and *Solyc12g005430.1*) were highly expressed in R12. *PAL2* and *CCR1* were highly expressed in R48. In R0_vs_R12, two metabolites (caffeic acid and sakuranetin) were downregulated. In R0_vs_R48, five metabolites (caffeic acid, sinapyl alcohol, 5-O-p-Coumaroylquinic acid*, 1-O-Sinapoyl-β-D-glucose, and coniferin) were downregulated. *CCR2* and *TOGT1* were highly expressed in S12, and *AnthOMT*, *HCT* (*Solyc12g088170.2*), *CHS1*, *CHS2*, *CAD6*, *PAL5*, *4CL1*, and *PER66* were highly expressed in S48. In S0_vs_S48, three metabolites (trans-5-O-(p-Coumaroyl)shikimate, isosalipurposide, and chlorogenic acid) were upregulated and four metabolites (L-Tyrosine, caffeic acid, 5-O-p-Coumaroylquinic acid*, and scopoletin) were downregulated (Figure 5A,B).

#### 2.4.2. KEGG Enrichment Analysis of Especially Expressed DEGs and DAMs

In order to study the unique information of resistant and susceptible materials inoculated with *O. neolycopersici* at each time point, we performed KEGG enrichment analysis of DEGs and DAMs specifically expressed only in R0_vs_R12, R0_vs_R48, S0_vs_S12, and S0_vs_S48, respectively. It was found that only in R0_vs_R12 were there 790 DEGs and seven pathways were notably enriched, including plant–pathogen interaction, sulfur metabolism, MAPK signaling pathway–plant, and so on. Only in R0_vs_R48 were there 266 DEGs and 4 pathways; processing in endoplasmic reticulum, monoterpenoid biosynthesis, biosynthesis of unsaturated fatty acids, and ribosome biogenesis in eukaryotes were significantly enriched. Only in S0_vs_S12 were there 1364 DEGs specifically expressed and 13 significantly enriched pathways including glycerophospholipid metabolism, riboflavin metabolism, folate biosynthesis, and so on. Only in S0_vs_S48 were there 739 DEGs specifically expressed and 12 significantly enriched pathways including other glycan degradation, glutathione metabolism, starch, and sucrose metabolism, and so on (Appendix A). Only in R0_vs_R12 were there 27 special accumulated metabolites and 3 significantly enriched pathways, namely, oxidative phosphorylation, glycine, serine, and threonine metabolism, and phenylalanine metabolism. Only in R0_vs_R48 were there 76 special accumulated metabolites, and 5 pathways with significant enrichment were involved in galactose metabolism, linoleic acid metabolism, and biosynthesis of unsaturated fatty acids. Only in S0_vs_S12 were there 52 special accumulated metabolites and 1 significantly enriched pathway, which was flavonoid biosynthesis. Only in S0_vs_S48 were there 76 special metabolites accumulated, and 5 pathways with significant enrichment were involved, including cyanoamino acid metabolism, aminoacyl–tRNA biosynthesis, and valine, leucine and isoleucine degradation (Appendix A). Among these results, we found that the biosynthesis of unsaturated fatty acids was a pathway in which the specific DEGs and DAMs of PM-resistant tomato leaves were significantly enriched at 48 hpi; two genes involved in this pathway were screened as *FAD2* (*Solyc12g049030.1* and *Solyc12g100260.1*), and four metabolites were α-linolenic acid, γ-linolenic acid, eicosadienoic acid, and linoleic acid. The pathways, genes, and metabolites obtained in the above studies may be the reason for the disease resistance of the material ‘63187’.

## 3. Discussion

*O. neolycopersici* is one of the most widespread and severe fungal diseases of tomato, impacting the growth and yield of tomato. Cultivating disease-resistant cultivars is the most economical and effective approach. However, it requires the knowledge of the interaction of tomato with *O. neolycopersici* and the mining of resistance genes from tomato. In this study, transcriptome and metabolome were used to analyze three different inoculation time points of two different resistant tomato leaves. This joint analysis will offer a uniquely promising chance to find out about the candidate genes and metabolites participating in the tomato disease resistance pathway.

Genes of the same type have similar trends of variation and may have similar functions under different experimental treatments [19]. In this study, K-means clustering analysis revealed seven genes with consistently increased expression in ‘63187’, and nineteen genes had consistently increased expression in ‘MM’, screening out seven possible candidate genes for disease resistance and nineteen candidate genes for susceptibility (Figure 5C). The expression of *AnthOMT* at 12 hpi was close to 0 hpi expression in ‘63187’, while it was 5.03-fold that of 0 hpi in ‘MM’; at 48 hpi, its expression was 0.36-fold that of 0 hpi in ‘63187’, while 7.02-fold that of 0 hpi in ‘MM’. Therefore, as a possible candidate gene for susceptibility, this gene needs further study. The continuous increase in *SGT1* expression in ‘63187’ may enhance the PM resistance of this material. Guo et al. [20] showed that over-expression of *CmSGT1* in pumpkin improved PM resistance. Xing et al. [21] silenced the *Hv-SGT1* gene reduced PM resistance in wheat, suggesting that *Hv-SGT1* was an integral part of disease resistance. It has been reported that after barley was infected with fungal PM, the activity of polyamine oxidase (PAO) related to the production of hydrogen peroxide increased, which may promote the defense of plants against the pathogen [22]. Xu et al. [23] found that compared to wild-type leaves, grape leaves with transient over-expression of *VdMYB1* showed a smaller number of fungal conidia, which increased the resistance of grape PM. In the present study, *PAO2* and *R2R3MYB* expressions were consistently elevated in ‘63187’ after inoculation with *O. neolycopersici* and could be used as a resistant candidate gene for further functional validation. No relationship between *PLAT1* and plant fungus has been found, but numerous studies suggested that it plays an enormous part in abiotic stresses and that knocking out *plat1* increases plant colonization [24]. In this study, the persistently elevated expression of *PLAT1* in ‘MM’ may have increased the susceptibility of this material to PM. Over-expression of the *RMA1H1* gene-enhanced drought tolerance in transgenic *Arabidopsis thaliana* is frequently quickly evoked under a range of abiotic stresses, but the response to biotic stresses has rarely been reported. The *RMA1H1* gene screened in this study will provide a research foundation for biotic stresses [25].

TFs are crucial in plant defense; they regulate gene transcription at a specific time and in a specific process and can coordinate the expression of many genes involved in plant defense [26]. In this study, 9855 DEGs were screened through the transcriptome and 796 TFs were annotated, which were divided into 80 transcription families. Among them, the first three families with more TFs were bHLH, AP2/ERF-ERF, and WRKY, which may play important roles in the infection of tomato leaves to *O. neolycopersici*. Biotic stress in plants activates salicylic acid (SA), jasmonic acid (JA), and ethylene (EI) signaling pathways, and then changes the transcription level of related genes and protein post-processing, responding to different biotic stresses [27]. The WRKY family is among the greatest families of specific TFs that perform a role in plant growth, development, stress, and especially disease resistance. It has been found that the over-expression of *VqWRKY31* increased resistance to PM in grape by promoting salicylic acid signal transduction and specific metabolite synthesis [28]. *AtWRKY33* and two homologous genes of tomato, *SlWRKY31*, and *SlWRKY33*, were activators of plant responses to several pathogens [29]; the silencing of the *SlWRKY23* gene resulted in increased resistance to PM but decreased resistance to salt stress, indicating that the response effects of plants to single stress might not be additive under multiple stresses [30]. In this study, by further analysis of the WRKY family, four TFs showed increased expression in ‘63187’ and one TF in ‘MM’. *Arabidopsis thaliana WRKY46*, *WRKY70*, and *WRKY53* were upregulated basal resistance to *pseudomonas syringae* [31]. *TaWRKY70* was involved in the resistance of wheat high-temperature seedlings (HTSP) to *Puccinia striiformis* f. sp. *Tritici* (*Pst*) induces stripe rust in wheat (*Triticum aestivum*), during which SA and ET signals may be activated. The silencing of *TaWRKY70* led to greater susceptibility to *Pst* [32]. In this study, the expression of *WRKY46* and *WRKY70* increased in ‘63187’ after inoculation with *O. neolycopersici* and there was no significant change in ‘MM’. It showed that both genes likely contribute to the resistance response of tomato leaves to PM, and may have a positive regulatory effect on resistance to PM.

In order to eliminate the gene differential expression caused by the genetic background of different materials, Zheng et al. [11] identified 5478 genes between resistant and susceptible cucumbers to PM. They analyzed of the functions of DEGs and evaluated the complex regulatory network of PM resistance according to GO and KEGG databases, involving several pathways that may be related to PM, such as phytohormone signaling, phenylpropanoid biosynthesis, plant–pathogen interactions and MAPK signaling pathways. Huang et al. [33] analyzed and compared the transcriptomic profiles of resistant and susceptible *Pueraria lobata* pseudo-rust disease, identified 7044 DEGs, and screened 406 co-expressed genes. It was further speculated that the response of *Pueraria lobata* to *Synchytrium puerariae Miy* infection may be related to oxidation reduction processes, flavonoid biosynthesis, and ABA signaling genes. A total of 9855 DEGs were obtained in this study. We selected 276 genes co-expressed by all comparison groups for KEGG analysis; of these, the pathways markedly enriched were amino acid metabolism, signal transduction, energy metabolism, and the synthesis of other secondary metabolites, and the changes of this transcriptome were related to the data of metabolic groups. Using the Pearson correlation algorithm to calculate correlations between the candidate metabolites and candidate genes, we found that the candidate genes were negatively correlated with lysoPE 16:3 and positively correlated with other candidate metabolites. The relative content of the seven and three metabolites screened in this study was consistently elevated in ‘63187’ and ‘MM’, respectively. In plants, there is a class of antitoxin compounds used to defend against biotic or abiotic stresses, the accumulation of which is important in pathogenic fungi-invading plants. The compounds are usually terpenoids, phenols, and alkaloids, whereas flavonoids are a large class of phenolic compounds, derived from a branch of phenylpropanoid biosynthesis. Thus, flavonoids, terpenoids, and alkaloids may have an important role in tomato leaves after inoculation with *O. neolycopersici*. Among the seven metabolites screened in ‘63187’ with continuously increasing relative content, there were three terpenoids (alphitolic acid, 2α-Hydroxyursolic acid, and 3,24-Dihydroxy-17,21-semiacetal-12(13)oleanolic fruit), one alkaloid ((22R,25R)-16β-H-22a-N-spirool-3β-ol-5-en-3-O-rhamnosyl(1→2)[rhamnosyl(1→4)]glucoside), and one flavonoid (Tricin-4′-O-glucoside-7-O-glucoside). Among the three metabolites screened in ‘MM’ with continuously increasing relative content, there were two flavonoids (isorhamnetin-3-O-(6″-O-p-coumaroyl)thapsigargin-7-O-rhamnoside and lignan-7-O-rutinoside-5-O-rhamnoside). 2α-Hydroxyursolic acid is a pentacyclic triterpenoid with reported anti-fungal and anti-bacterial activity [34]; alphitolic acid is a compound with anti-inflammatory activity and inhibits Gram-positive bacteria, anti-HIV activity [35]; through the isolation of oleanolane-type *Triterpenoid saponins* from the pulp extract of sapindus, it was found to have anti-fungal activity [36]. Therefore, the relative content of the three metabolites screened in ‘63187’ may increase the resistance of the material to PM, which was consistent with the anti-bacterial activity of the metabolite in previous studies. The combined analysis of transcriptome and metabolome revealed that phenylpropane biosynthesis and flavonoid biosynthesis were both enriched in the disease-resistant tomato inbred line ‘63187’ and the susceptible tomato variety ‘MM’ (Figure 5A,B). A study found that many flavonoid compounds in cucumber-resistant varieties were upregulated after infection with PM, which suggested that the increase in flavonoids was crucial to the resistance of cucumber to PM [37]. Yu et al. [38], through extensive targeted metabolomic analysis, found that the defense mechanism of grapefruits against PM was mainly related to phenylpropane-flavonoid metabolism, which was compatible with our research results.

Former studies have shown that phenylalanine ammonia lyase (PAL) was an important enzyme related to the defense of phenylpropane pathway, involving a secondary metabolism related to defense, and the strengthening of the cell wall may confer disease resistance to tomato [39]. *PAL1* increased the resistance of *cassava brown streak virus* [40]. After the knock-down of *Brachypodium distachyon PAL1* (*BdPAL1*) by RNA interference (RNAi), virus-induced SA and lignin accumulation were significantly inhibited, increasing the susceptibility to TMV. Therefore, the *PAL1* gene played an important role in resisting viruses, while *PAL5* was mainly responsible for cell wall strengthening and the provision of chemical precursors for phenolic compounds of anti-nutritional compounds [41]. In this study, the *PAL2* gene was highly expressed in R48 and the *PAL5* gene was highly expressed in S48, so they could be used as PM candidate genes for functional studies. Linolenic acid is a fatty acid present in the lipid membrane and a precursor of the lipid oxidation pathway that leads to jasmonic acid. Therefore, the accumulation of linolenic acid played a very important role in resisting plant diseases and insect pests [42]. Linoleic acid is a source of various oxidative metabolites, called oxidized lipids, some of which can inhibit fungal pathogens, and the accumulation of hydroxylinoleic acid has the role of enhancing the ability of rice fungus [43]. Eicosadienoic acid accumulation contributed to enhancing abscisic acid sensitivity to mitigate drought effects in transgenic *Arabidopsis thaliana* [44]. It has been reported that silencing the *SlFAD2-7* gene in suppression of *prosystemin-mediated responses 2* (*spr2*) resulted in an increase in aphid population, indicating that *SlFAD2-7* helped to improve aphid resistance. The *FAD2* tomato gene family was vital for both primary fatty acid metabolism and the response to biological stress [45]. In this study, it was found that linolenic acid, linoleic acid, and eicosadienoic acid accumulated in ‘63187’ at 48 hpi, and the high expression of *FAD2* in ‘63187’, may be one of the reasons leading to the disease resistance of ‘63187’, which was consistent with previous research results. The candidate genes obtained from the above research can be further validated for their functions through gene over-expression, gene knockout, gene silencing, and other methods to obtain transgenic plants and cultivate disease-resistant varieties. Materials with resistance genes can also be used as parents, and materials can be created through backcrossing, hybridization, and other methods to improve the parents and gradually obtain resistant varieties.

## 4. Materials and Methods

### 4.1. Plant Growth, Inoculation, and Propagation of O. neolycopersici

The PM-resistant tomato inbred line, ‘63187’, and the susceptible tomato variety, ‘MM’, with full seeds were sown in 98-hole plugs, and the temperature was controlled at 25–28 °C by day and 18–20 °C by night in the plant growth chamber. When the seedlings grew from cotyledon to two leaves and one heart leaf, the seedlings were transplanted into flowerpots and continued to grow. When the seedlings in the flowerpot grow to four leaves and one heart leaf, the purified *O. neolycopersici* was mixed with sterile water to prepare a 1 × 10^6^/mL spore suspension, which was sprayed evenly on the tomato leaves with a watering can [46]. The control tomato leaves were sprayed with the same amount of clear sterile water. The temperature and the relative humidity were controlled at 27 °C and 80% during the day, while they were 20 °C and 75% at night. The photoperiod was L//D = 12 h//12 h [47]; samples were collected at 0 hpi, 12 hpi, and 48 hpi, and stored in a freezer tube in a −80 °C refrigerator. There were three biological replicates at each time point after inoculation. Each biological replicate contained three plants, and the first true leaf was collected from each plant.

### 4.2. RNA Extraction and RNA-Seq

The total RNA from disease-resistant and susceptible tomato leaves were extracted with the TIANGEN RNA kit (TIANGEN, Beijing, China). The detection of RNA integrity of samples was conducted by garose gel electrophoresis. We used a NanoPhotometer spectrophotometer (IMPLEN, Los Angeles, CA, USA) to detect the RNA sample’s purity (OD_260_/_280_ and OD_260_/_230_ ratio). Precise quantification of the RNA concentration was conducted with the Qubit2.0 Fluorometer (Life Technologies, Carlsbad, CA, USA). Precise detection of the RNA integrity was conducted with the Agilent2100 bioanalyzer (Agilent Technologies, Santa Clara, CA, USA). Using the total RNA with an amount of ≥1 μg as a template, building cDNA libraries was conducted with the Illumina NEBNext^®^ UltraTM RNA Library Prep Kit (NEB, Ipswich, MA, USA). In total, 18 RNA-Seq libraries, namely, 6 treatments consisting of uninoculated and inoculated plants of the resistant and susceptible lines (R0, R12, R48, S0, S12, and S48) with 3 replications of each combination, were separately constructed. Initial quantification of the constructed library was performed using Qubit2.0 and the insert size of the library was tested using Agilent2100. After the test was qualified, use the Illumina platform for sequencing (San Diego, CA, USA).

### 4.3. Read Mapping and Data Analysis

In order to obtain high-quality reads, Fastp was employed to strictly control the data, i.e., to eliminate reads with adapters, low-quality reads with N bases exceeding 10% of the reads, and those with more than 50% Qphred ≤ 20 bases [48]. We calculated the GC content of clean reads and used FastQC to calculate the values of Q20 and Q30 to evaluate base quality. Using HISAT2 and default parameters to map clean reads to the tomato reference genome, fragments per kilobase per million fragments-mapped transcripts (FPKM) were used as a marker for transcripts or gene expression levels [49]. This process used feature counts [50]. Correlations between the three biological replicates were assessed by applying Pearson’s correlation coefficient (R). R^2^ was at least greater than 0.8 and closer to 1, indicating that the correlation between samples was good [51]. DEGs were screened using DESeq2 [52,53] software with |log_2_fold change (FC)| ≥ 1 and false discovery rate (FDR) < 0.05 as criteria. The function of DEGs was investigated using Gene Ontology (GO) functional analysis and Kyoto Encyclopedia of Genes and Genomes (KEGG) [54] pathway annotation. GO is an international classification of gene function into cellular components, biological processes, and molecular functions, which is used to define and describe gene and protein functions in each species [55]. The scale function of R language was used to standardize the FPKM values of the differential genes, and K-means clustering analysis was conducted. Using iTAK software for plant transcription factor prediction [56], iTAK incorporates two databases, PlnTFDB, and PlantTFDB [57,58].

### 4.4. Validation of the Quantitative PCR

For the expression validation, we selected nine candidate genes for quantitative real-time PCR analysis, with the *Act* gene serving as a reference gene [59]. We designed specific primers using Premier 5.0, shown in Appendix A. RNA extraction was performed using a TIANGEN kit (TIANGEN, Beijing, China), reverse transcription kits (Vazyme, Nanjing, China) were deployed to reverse transcribe the purified RNA (1 μg per sample) into first-strand cDNA, and real-time fluorescence quantification was performed using a Qtower2.0 quantitative PCR instrument (Analytik Jena, Jena, Thurin-119 gia, Germany) using one biological replicate; that is, three leaves were used to extract one RNA sample and three technical duplicates, and quantitative assessment was undertaken via the 2^−ΔΔCt^ method [60].

### 4.5. Metabolite Extraction

Freeze drying of biological samples using a vacuum freeze dryer (Scientz-100F). A zirconia bead was added to the freeze-dried sample, which was continuously ground for 1.5 min at 30 Hz using a grinder (MM 400, Retsch). The 50 mg of lyophilized powder was dissolved in 1.2 mL of 70% methanol solution and vortexed for 30 s at 30-min intervals for a total of 6 times, then centrifuged (12,000 rpm for 3 min). The supernatant was aspirated, and the sample was filtered through a microporous membrane (0.2 µm pore size) and stored for analysis by UPLC-MS/MS in a sample vial [61].

### 4.6. Qualitative and Quantitative Metabolite Analysis

The processing of mass spectrometry data was carried out using the software Analyst 1.6.3. The qualitative analysis of metabolites using secondary spectral information based on the self-built database MWDB (metware database) and the multiple reaction monitoring (MRM) modes of triple quadrupole mass spectrometry was used for the quantitative analysis of metabolites. Principal component analysis (PCA) plots were used to visualize between- and within-group variability. In supervised Orthogonal Partial Least Squares Discriminant Analysis (OPLS-DA), R^2^Y and Q^2^ are the forecast indicators for the evaluation model. Generally, Q^2^ > 0.5 is a valid model, and Q^2^ > 0.9 is an outstanding model [62]. For the data from OPLS-DA, combined with the variable importance projection (VIP) and FC value of the OPLS-DA model, the VIP values ≥ 1, |log_2_FC| ≥ 1 were used as the thresholds for screening differential metabolites. The relative content of 46 DAMs was treated with unit variance (UV), and then K-means clustering analysis was carried out. The KEGG database was used for the functional annotation of DAMs [63].

### 4.7. Combined Transcriptome and Metabolome Analysis

During the joint analysis of transcriptome and metabolome, Pearson correlation coefficients of genes and metabolites were calculated using the cor function in R. Correlations with correlation coefficients greater than 0.80 and *p*-value less than 0.05 were selected and correlation networks were plotted using Cytoscape software [64]. The differential metabolites and differential genes in all treatments were mapped to their associated KEGG pathways for the purpose of better understanding the relationship between genes and metabolites.

## 5. Conclusions

In this study, 9855 DEGs were identified by transcriptome sequencing and 276 genes were co-expressed in all treatments, screening out 7 possible candidate genes for disease resistance and 19 possible candidate genes for susceptibility. The analysis of WRKY family transcription factors identified four possible candidate TFs for resistance to PM and one possible candidate TF for susceptibility to PM. In addition, 1058 DAMs were detected, and 46 DAMs were co-accumulated in all treatments, screening out 8 possible candidate metabolites for disease resistance and 3 possible candidate metabolites for susceptibility. We found that the accumulation of flavonoids and terpenoids may respond positively to tomato leaves’ inoculation with *O. neolycopersici*. We combined transcriptome and metabolome data and found that both phenylpropanoid biosynthesis and flavonoid biosynthesis pathways were enriched in ‘63187’ and ‘MM’. The biosynthesis of unsaturated fatty acids was the pathway through which the specific DEGs and DAMs of PM-resistant tomato leaves were significantly enriched at 48 hpi. These results provide favorable molecular information for the study of different resistances of tomatoes to PM, and they provide a foundation for the breeding of tomato varieties resistant to PM.

## Figures and Tables

**Figure 1 ijms-24-08236-f001:**
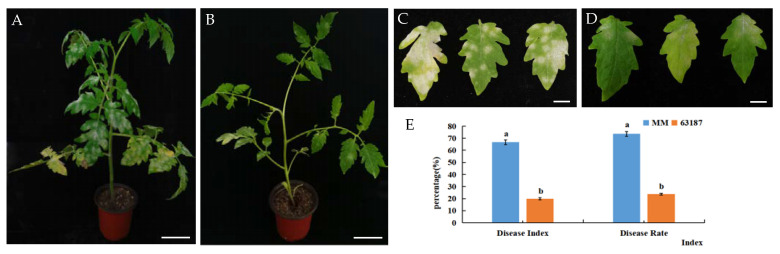
Phenotypes of ‘MM’ and ‘63187’ after inoculation with *O. neolycopersici* 25 d. (**A**,**B**) Phenotypes for ‘MM’ and ‘63187’ tomato plants after inoculation with *O. neolycopersici* 25 d, with a scale of 1 cm. (**C**,**D**) Phenotypes for ‘MM’ and ‘63187’ tomato leaves after inoculation with *O. neolycopersici* 25 d, with a scale of 0.5 cm. (**E**) Histograms of disease index and incidence rate for ‘MM’ and ‘63187’. Lower case letters denote discrepancy at the 0.05 level.

**Figure 2 ijms-24-08236-f002:**
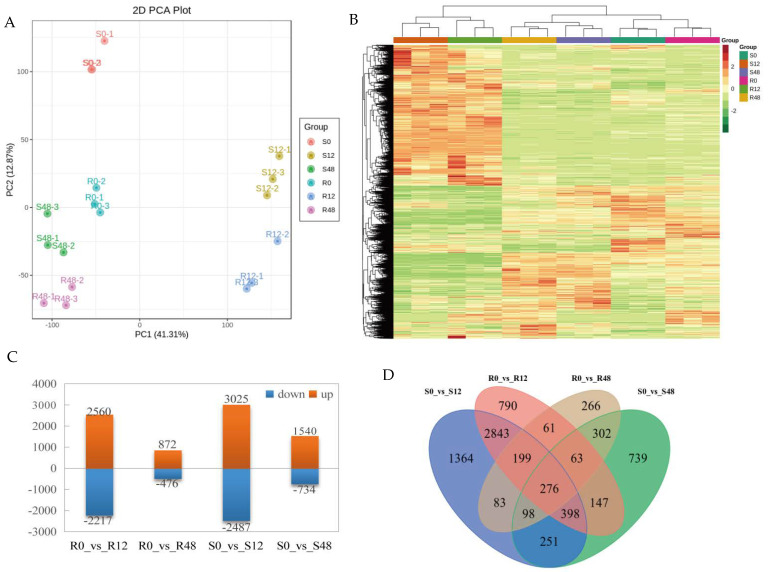
An overview of RNA-seq data. (**A**) Principal component analysis (PCA) among samples. (**B**) Level clustering heat map. The level axis stands for the sample name and level clustering results; the longitudinal axis stands for DEGs and level clustering results; red and green, respectively, mean high and low expression. (**C**) Distribution of DEGs at distinct time points in resistant and susceptible tomato plants after *O. neolycopersici* treatment. (**D**) A Venn diagram showing the DEGs numbers in each comparison group.

**Figure 3 ijms-24-08236-f003:**
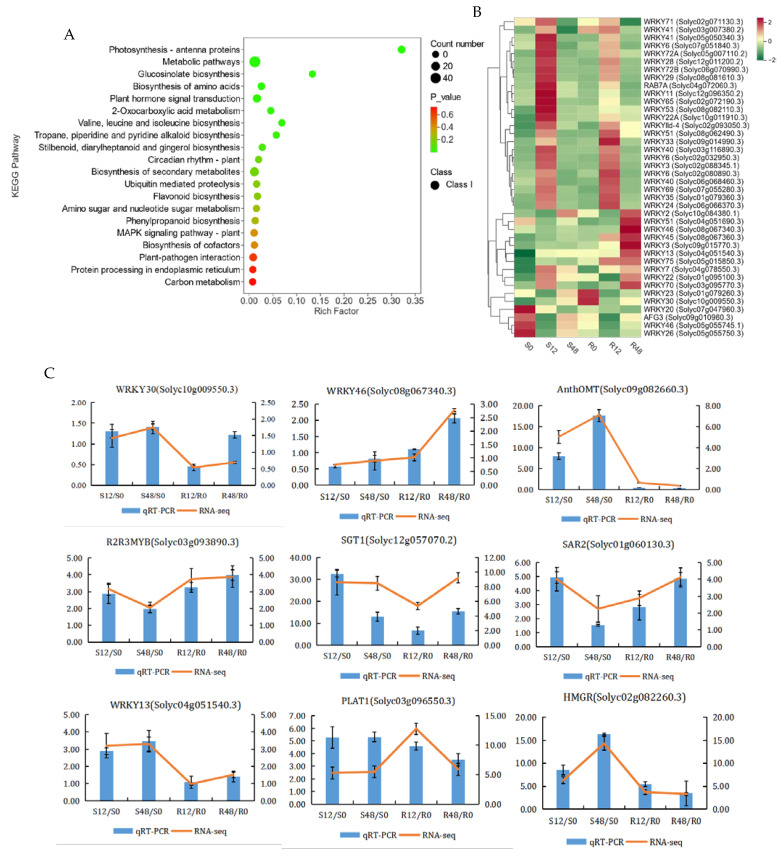
Enrichment analysis and qRT-PCR validation of DEGs and WRKY analysis. (**A**) KEGG enrichment scatter plot of DEGs in the four compared groups. The vertical axis denotes the KEGG pathway, and the horizontal axis denotes the rich factor. The greater the rich factor, the greater the enrichment degree. The larger the dot, the higher amount of differentially enriched genes by the pathway. The smaller the P-value, the more significant the enrichment degree. (**B**) WRKY family genes expression analysis. The horizontal axis represents the different sample names, and the vertical axis represents DEGs. Red and green mean high and low expression, respectively. (**C**) Validation of DEGs by qRT-PCR analysis. The bar graphs represent the qRT-PCR data; the line plots represent the RNA-Seq data.

**Figure 4 ijms-24-08236-f004:**
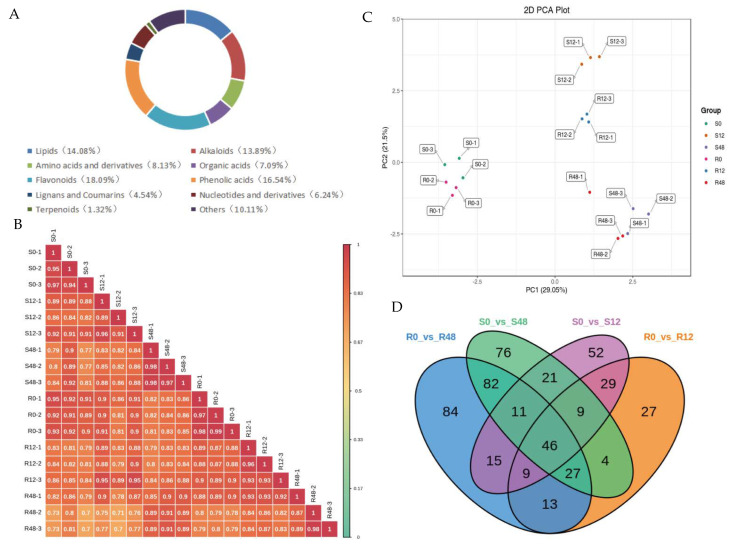
Qualitative and quantitative analysis of metabolome data. (**A**) Metabolite categories make up a ring diagram. Each color represents a metabolite category, and the area of the color block indicates the proportion of the category. (**B**) Correlation analysis between samples. The vertical and diagonal lines represent the sample names, and different colors represent different Pearson correlation coefficients. (**C**) PC1 and PC2 are the first and second principal components, correspondingly, and the percentage means the explanation rate of the principal component to the data set. Every point in the graph indicates a sample, and samples from the same group are marked with the same color. (**D**) A Venn diagram of DAMs in four comparison groups.

**Figure 5 ijms-24-08236-f005:**
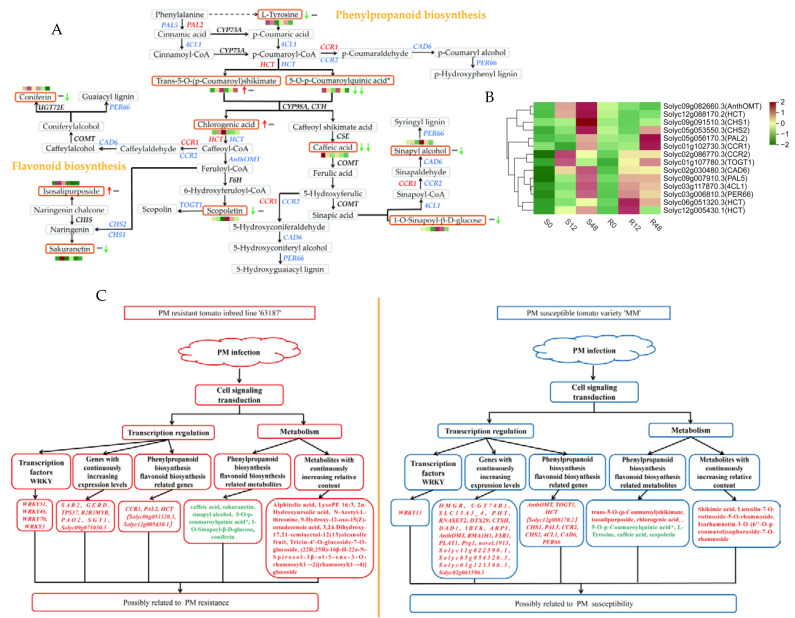
Metabolome changes and transcriptome regulation of tomato leaves with different resistance after inoculation with *O. neolycopersici*. (**A**) A network diagram of phenylpropane biosynthesis and flavonoid biosynthesis pathways of resistant tomato inbred line, ‘63187’, and susceptible tomato variety, ‘MM’, after inoculation with *O. neolycopersici*; red font represents possible resistance candidate genes, blue font represents possible susceptibility candidate genes, the gray rectangular box indicates the relative content of metabolites remains unchanged, the orange rectangular box represents that the relative content of metabolites has changed, red and green arrows represent upregulated and downregulated metabolites, respectively, and gray horizontal lines represent the unchanged relative content of metabolites. The arrows or horizontal lines right to the rectangular boxes represent the metabolites accumulation trend in ‘MM’ on the left and ‘63187’ on the right, respectively. (**B**) Possible candidate gene heat map. The rectangular squares from left to right represent six samples, S0, S12, S48, R0, R12, and R48; red denotes high expression and green denotes low expression. (**C**) Summary map of the PM-resistant and the susceptible. The orange line in the middle is taken as the dividing line, the left side represents the PM-resistant summary map, the right side represents the PM-susceptible summary map, the green font is the downregulated metabolites, and the red font is the upregulated metabolites and genes.

## Data Availability

We guarantee the authenticity of all data in this study.

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
