# Peer review of "Comparative Transcriptome and Widely Targeted Metabolome Analysis Reveals the Molecular Mechanism of Powdery Mildew Resistance in Tomato"

_ijms, 2023, doi:10.3390/ijms24098236_

Round 1

Reviewer 1 Report

ijms-2335488 is very interesting and give the valuable information to the researchers and readers. The subject of the manuscript is consistent with the scope of the Journal, I suggested that the manuscript need to be minor revised before it is accepted by this journal.

1. Line 97: this sentence has grammatical problem.

2. Line 113: delete “Using”.

3. Line 143: the full name of “FDR” must be written where the abbreviations are first used.

4. Line 320, 327, 334 and 342: delete “Data” in “Supplementary Data Table”.

5. Line 328 and 335: set the “2” in “log2FC” to a subscript.

6. Line 572-574: please change the “pueraria lobata” and “Synchytrium puerariae Miy” to italics, and some words require capital letters.

7. Line 649-655: please add the manufacturer and origin of the kits and instruments used.

8. Line 653 and 676: replace “ug” with “μg”.

9. Line 674: replace “Sheet” with “Table”.

10. Check for missing and redundant spaces in the text.

11. Check the format of all references in list.

Author Response

Response to Independent Review Report, Reviewer 1

We would like to thank the reviewer for the careful and constructive reviews. Based the comments from the reviewer, we have revised the manuscript, which are detailed below.

  1. Line 97: this sentence has grammatical problem.

Response 1: This sentence has been modified to remove the word "In". (Please see line 101 of the revised manuscript)

  1. Line 113: delete “Using”.

Response 2: OK, we have deleted “Using” . (Please see line 117 of the revised manuscript)

  1. Line 143: the full name of “FDR” must be written where the abbreviations are first used.

Response 3: We moved this sentence to the Materials and Methods section based on the feedback of another reviewer, and used its full name when it first appeared according to the reviewer's feedback. (Please see line 661 of the revised manuscript)

  1. Line 320, 327, 334 and 342: delete “Data” in “Supplementary Data Table”.

Response 4: According to the reviewer's comments, the "Data" in lines 320, 327, 334, and 342 of the "Supplementary Data Table" has been deleted. (Please see lines 320, 328, 336, and 343 of the revised manuscript)

  1. Line 328 and 335: set the “2” in “log2FC” to a subscript.

Response 5: According to the reviewer's comments, “2” in “log2FC” has been set as the subscript in lines 328 and 335. (Please see lines 329 and 337 of the revised manuscript)

  1. Line 572-574: please change the “pueraria lobata”and “Synchytrium puerariae Miy” to italics, and some words require capital letters.

Response 6: The words in lines 572-574 that require italics and uppercase letters have been modified. (Please see lines 543 and 545 of the revised manuscript)

  1. Line 649-655: please add the manufacturer and origin of the kits and instruments used.

Response 7: The manufacturer and source of the tool kit and instruments used in lines 649-655 have been added. (Please see lines 634-648 of the revised manuscript)

  1. Line 653 and 676: replace “ug” with “μg”.

Response 8: Replaced “ug” in lines 653 and 676 with “μg”. (Please see lines 641 and 676 of the revised manuscript)

  1. Line 674: replace “Sheet”with “Table”.

Response 9: Line 674 “Sheet” has been replaced with “Table”. (Please see lines 673 of the revised manuscript)

  1. Check for missing and redundant spaces in the text.

Response 10: Checked for missing and redundant spaces in the entire text.

  1. Check the format of all references in list.

Response 11: The format of the references in the article has been checked and modified.

Reviewer 2 Report

Language needs to improve. It is unacceptable in current format.

Format graph 1E X -axis ligand properly.

Use same notation of powdery mildew in text and figure.

How GC content is relevant to sequence quality

Each section of result start with kind of description of method ( 161-168;141-143 etc)  that should be part of method.

Mention and describe, how many genes and what genes were differentially express in in resistant cultivar in  comparison susceptible at 12 and 14 hpi on both transcriptome and metabolic analysis.

Move figure 3 C-d to supplementary. In q-PCR compare expression between susceptible and resistance line.

Move figure 5 to supplementary information.

Elaborate section 2.2.5 on q-PCR and its relevance with corresponding RNAseq results.

Figure 6 C-D needs to be created with pathway that up or down regulated in suspectable and resistance line of tomato.

For easy interpretation make a table with DEG and DAM genes that are mentioned in section 2.4 and 2.5.

Section 4.1 and 2 provide details of biological and technical replicates.

Discussion needs to be shortened with focus on resistance genes and their pathway for further incorporation in breeding.

Conclusion needs to be re-written with emphasis on genes that may be related to resistance to PM in tomato.

Other general comment:

Line 37: what do you mean global fungal disease

Line 80: resistance mechanism  of PM is well described by many authors. Include these information. One to these article is: https://www.ncbi.nlm.nih.gov/pmc/articles/PMC6640449/

Line 87:Remove “with 0 hpi as control (CK)”

Line 123: How GC content is relevant to sequence quality

Line no 145-146: rephrase

Author Response

Response to Independent Review Report, Reviewer 2

We would like to thank the reviewer for the careful and constructive reviews. Based the comments from the reviewer, we have revised the manuscript, which are detailed below.

  1. Language needs to improve. It is unacceptable in current format.

Response 1: The language section has been revised at https://www.mdpi.com/authors/english, with an English editorial ID of english-64823. Here is the proof of modification.

  1. Format graph 1E X -axis ligand properly.

Response 2: The X -axis of figure 1-E has been correctly set to “Index”. (Please see line 110 of the revised manuscript)

  1. Use same notation of powdery mildew in text and figure.

Response 3: In this manuscript, “Powdery mildew” in both text and figures are represented by “PM”, and powdery mildew fungus in this manuscript is “Oidium neolycopersici, so we use O. neolycopersici” as notation.

  1. How GC content is relevant to sequence quality.

Response 4: There is no direct correlation between GC content and sequencing quality. The purpose of this sentence is only to describe the CG content. We are very sorry, this sentence may not have appeared in the right place, we have moved it to the right place. (Please see line 127 of the revised manuscript). Q20 and Q30 correlate with sequencing quality. Q20 and Q30 represent the percentages of bases whose accuracy was above 99% and 99.9%, respectively. In this study, Q20 base was about 97.24%, and the Q30 base was about 91.91%, suggesting the relatively high quality of transcriptome data.

  1. Each section of result start with kind of description of method (161-168;141-143 etc) that should be part of method.

Response 5: The methods appearing in the test results have been moved to the Materials and Methods section. (Please see lines 663-665 and 660-661 of the revised manuscript)

  1. Mention and describe, how many genes and what genes were differentially express in in resistant cultivar in comparison susceptible at 12 and 14hpi on both transcriptome and metabolic analysis.

Response 6: There may be background differences between disease-resistant and susceptible materials, and research has shown that using co-expressed genes and metabolites can eliminate such differences [1-3]. Therefore, in this article, we mainly analyzed 276 differentially expressed genes (DEGs) and 46 differentially accumulated metabolites (DAMs) shared by both resistant and sensitive materials. DEGs were mainly analyzed through transcriptome. Metabolome can only be used to screen metabolites, and there are many genes controlling metabolites. Therefore, we mainly use enriched metabolic pathways to conduct joint analysis on them, and at the same time, we conduct correlation analysis on candidate genes and candidate metabolites screened by transcriptome and metabolome. Of course, in addition to co-expressed genes, there are also many differentially expressed genes, and there is a lot of information that omics can mine. Due to space limitations in the article, we are considering delving deeper into the data according to your suggestions and writing another article.

[1] Yang, C.C.; Wu, P.F.; Yao, X.H.; Sheng, Y.; Zhang, C.C.; Lin, P.; Wang, K.L. Integrated transcriptome and metabolome analysis reveals key metabolites involved in camellia oleifera defense against anthracnose. Int J Mol Sci. 2022, 23, 536.

[2] Zhao, F.; Niu, K.; Tian, X.; Du W. Triticale improvement: mining of genes related to yellow rust resistance in triticale based on transcriptome sequencing. Front. Plant Sci. 2022, 13, 883147.

[3] Yuan, H.; Zeng, X.; Yang, Q.; Xu, Q.; Wang, Y.; Jabu, D.; Sang, Z.; Tashi, N. Gene co-expression network analysis combined with metabonomics reveals the resistance responses to powdery mildew in Tibetan hulless barley. Sci. Rep. 2018, 8, 14928.

  1. Move figure 3 C-d to supplementary. In q-PCR compare expression between susceptible and resistance line.

Response 7: Figure 3 C-d has been moved to the supplementary figure. (Please see figure S3 of supplementary files). The expression of sensitive and resistant lines in qRT-PCR has been added to 2.2.5. (Please see lines 238-256 of the revised manuscript)

  1. Move figure 5 to supplementary information.

Response 8: Figure 5 has been moved to supplementary information. (Please see figure S9 of supplementary files)

  1. Elaborate section 2.2.5 on q-PCR and its relevance with corresponding RNAseq results.

Response 9: For details of q-PCR, please refer to 2.2.5, the graph of correlation coefficients has been supplemented to figure S6. (Please see lines 230-237 of the revised manuscript and figure S6 of supplementary files)

  1. Figure 6 C-D needs to be created with pathway that up or down regulated in suspectable and resistance line of tomato.

Response 10: We are very sorry for the trouble we have caused you in reading this due to our inappropriate language. The “network diagram” has now been revised to a “summary diagram”. The original figure 6C is now modified to figure 5C. This figure mainly summarizes the genes, metabolites, transcription factors, and metabolic pathways screened in the text, which was different from the meaning expressed in figure 5A. The orange line in the middle is taken as the dividing line, the left side represents the PM resistant summary map, the right side represents the PM susceptible summary map, the green font is the downregulated metabolites, and the red font is the upregulated metabolites and genes. In the next study, the disease resistance candidate genes on the left were mainly verified by over-expression and the disease susceptibility candidate genes on the right were mainly verified by silencing. Figure 5A constructs upregulated and downregulated genes and metabolites involved in phenylpropanoid biosynthesis and flavonoid biosynthesis pathways in resistant and susceptible tomatoes.

  1. For easy interpretation make a table with DEG and DAM genes that are mentioned in section 2.4 and 2.5.

Response 11: This table has been added. (Please see table S10 of supplementary files)

  1. Section 4.1 and 2 provide details of biological and technical replicates.

Response 12: The detailed information on biological and technological replication provided in sections 4.1 and 2 has been supplemented. (Please see lines 630-632 and 642-645 of the revised manuscript)

  1. Discussion needs to be shortened with focus on resistance genes and their pathway for further incorporation in breeding.

Response 13: This part of content has been added to discussion. (Please see the revisions in discussion)

  1. Conclusion needs to be re-written with emphasis on genes that may be related to resistance to PM in tomato.

Response 14: Conclusion section has been rewritten. (Please see the revisions in conclusion)

  1. Line 37: what do you mean global fungal disease

Response 15: This sentence may not be appropriate and has now been revised to “Tomato powdery mildew (PM) is a widely distributed and rapidly spreading fungal disease”. (Please see lines 38-39 of the revised manuscript)

  1. Line 80: resistance mechanism of PM is well described by many authors. Include these information. One to these article is: https://www.ncbi.nlm.nih.gov/pmc/articles/PMC6640449/

Response 16: There may be more research on the resistance mechanism of other species to PM, but there is relatively little research on the resistance mechanism of tomato PM. Therefore, we changed “the resistance mechanism of tomato to PM is largely unknown” to “there was relatively little research on the mechanism of powdery mildew resistance in tomatoes”. (Please see lines 84-85 of the revised manuscript)

  1. Line 87:Remove “with 0 hpi as control (CK)”.

Response 17: Okay, we have removed “with 0 hpi as control (CK)”. (Please see line 91 of the revised manuscript )

  1. Line 123: How GC content is relevant to sequence quality.

Response 18: This question is repeated with question 5. We have answered this question, please refer to response 5.

  1. Line no 145-146: rephrase.

Response 19: Lines 146-157 has been changed to “In 'MM', 3025 and 1540 DEGs were upregulated, and 2487 and 734 DEGs were downregulated at 12 hpi and 48 hpi, respectively”. (Please see lines 145 of the revised manuscript )

Reviewer 3 Report

My comments can be found in the attached MS.

Author Response

Response to Independent Review Report, Reviewer 3

We would like to thank the reviewer for the careful and constructive reviews. Based the comments from the reviewer, we have revised the manuscript, which are detailed below.

  1. Line 16, Could you please rewrite this one? Like.. powdery mildew is a serious problem in tomato production...

Response 1: Of course, we have changed this sentence to “Powdery mildew is a serious problem in tomato production”. (Please see line 16 of the revised manuscript)

  1. Line 38, Year?

Response 2: OK, we have modified it to “ It first broke out in Taiwan Province, China in 1919”. (Please see line 39 of the revised manuscript)

  1. Line 44, Is it in field or greenhouse?

Response 3: By reviewing the references, “in greehouse” has been added. (Please see line 45 of the revised manuscript)

  1. Line 46, Also powdery mildew control could be problematic in organic production of tomato, as most of the biologicals may not be effective.

Response 4: OK, we have added this sentence to the text based on your feedback. (Please see lines 48-49 of the revised manuscript)

  1. Change line 80 “of”to “in”.

Response 5: OK, we have changed “of” to “in”. (Please see line 85 of the revised manuscript)

  1. Line 92, would provide?

Response 6: Yes, “will give” has been changed to “would provide”. (Please see line 96 of the revised manuscript)

  1. Line 98, evaluated?

Response 7: Both incidence rate and disease index are “calculated”. To avoid repetition, disease index is “calculated”, incidence rate is “counted”, and disease resistance is “evaluated”. Therefore, the word has not been modified and is still used as “counted”.

  1. Line 99, what do you mean by diverse here? Was it resistant or moderately resistant?

Response 8: The meaning was that the two materials exhibit different levels of resistance. The classification criteria of group disease resistance are: immunity (I), disease resistance (R), moderate resistance (MR), susceptibility (S), and high sensitivity (HS). According to the classification criteria of group disease resistance, 'MM' was high-sensitivity (HS) and '63187' was resistant (R). 

  1. Line 112, Identification and...?

Response 9: I'm very sorry, this is an error we made in the format modification of our article. “Identification” should be deleted here. (Please see line 116 of the revised manuscript)

  1. Line134-135, This can go to the discussion part.

Response 10: Based on your feedback, this section is indeed not suitable for the results and analysis and we have moved it to Materials and Methods. (Please see lines 658-659 of the revised manuscript)

  1. Line 161-163, Does not belong here, may be in discussion or somewhere in introduction.

Response 11: OK, based on your feedback, this section is indeed not suitable for the results and analysis and we have moved it to Materials and Methods. (Please see lines 663-665 of the revised manuscript)

  1. Line 190-191, Belongs to discussion.

Response 12: Yes, this section indeed belongs to the discussion section, and we have moved it to the discussion section. (Please see lines 481-482 of the revised manuscript)

  1. Line 212-213, Belongs in discussion.

Response 13: Yes, this section indeed belongs to the discussion section, and we have moved it to the discussion section. (Please see lines 488-489 of the revised manuscript)

  1. Line 262, “different resistance”change “from MM and '63187'”.

Response 14: OK, we have been changed “different resistance” to “from MM and '63187'”. (Please see lines 268-269 of the revised manuscript)

  1. Line 290, “results according to”change “data from”

Response 15: This sentence has been moved to Materials and Methods, and “results according to” has been changed to “data from”. (Please see line 699 of the revised manuscript)

  1. Line 310, delet “of different samples”.

Response 16: OK, we have been deleted “of different samples”. (Please see line 310 of the revised manuscript)

  1. Line 384-385, different resistant menas 'MM' and '63187'?

Response 17: Yes, “different resistant tomato materials” means “'MM' and '63187'”.

  1. Line 389, insert a space between “processes”and “Pearson”.

Response 18: OK, we have been inserted a space between “processes” and “Pearson”. (Please see line 369 of the revised manuscript)

  1. Line 480, cultivar?

Response 19: Yes, '63187' is an inbred line that we have isolated, not a wild species, but a cultivar.

  1. Line 482, delet “is a fungal disease resulted by the PM pathogen and”.

Response 20: OK, we have been deleted “is a fungal disease resulted by the PM pathogen and”. (Please see line 473 of the revised manuscript)

  1. Line 483, add “fungal” between “severe” and “issues”.

Response 21: OK, “fungal” has been added between “severe” and “issues”. (Please see line 473 of the revised manuscript)

  1. Line 486, in the current study?

Response 22: We recommend “In this study” instead of “During this article”. (Please see line 477 of the revised manuscript)

  1. Line 490, defense?

Response 23: The discussion needs to be streamlined according to the opinion of another reviewer, so the content has been deleted. (Please see line 473 of the revised manuscript)

  1. Line 498, cultivars?

Response 24: The discussion needs to be streamlined according to the opinion of another reviewer, so the content has been deleted. (Please see line 473 of the revised manuscript)

  1. Line 573, italicize

Response 25: OK, Synchytrium purearia Miy” has been changed to italics. (Please see line 545 of the revised manuscript)

  1. Line 611, give the full genus here.

Response 26: Added the full name of Brachypodium distachyon. (Please see line 589 of the revised manuscript)

  1. Line 639, where was this experiment conducted? Greenhouse or growth chamber?

Response 27: This experiment was conducted in the plant growth chamber. This content has been added to the text. (Please see line 621 of the revised manuscript)

  1. Line 639, Spores? Spores of which powdery mildew species? In the lines 38 and 39, you mentioned several species.

Response 28: Inoculated with O. neolycopersici. This content has been added to the text. (Please see line 624 of the revised manuscript)

  1. Line 641, Please correct the tense here.

Response 29: OK, we have changed “spray” to “sprayed”. (Please see line 622 of the revised manuscript)

  1. Line 671, could you please provide information on biological and technical replicates?

Response 30: OK, detailed information has been added to the text, see 4.4 for details. (Please see lines 678-679 of the revised manuscript)

  1. Line 705, analyzed?

Response 31: According to another reviewer, the conclusion needs to be rewritten, so the word has been deleted. (Please see line 714 of the revised manuscript)

  1. Line 706, results indicated? In the conclusions usually you do not refer the pictures and citations.

Response 32: OK, we have deleted “Figure 6C” in conclusions. (Please see line 714 of the revised manuscript)

Round 2

Reviewer 2 Report

Thanks for improving significant part of MS. 

There is are few areas that still need to be addressed. 

1. Compare resistant and suspectable cultivars at 12 and 24 hpi and provide information on how many genes are expressed differently. These genes may be used as target genes for resistant breeding.

Provide at least one section on analysis.

2. Same kind of analysis for metabolite. 

These two analyses will provide genes and metabolites information that will be useful for PM resistance knowledge.  

Next time please provide a clean MS. You can indicate change in your reply.

Author Response

We would like to thank the reviewer for the careful and constructive reviews. Based the comments from the reviewer, we have revised the manuscript, which are detailed below.

  1. Compare resistant and suspectable cultivars at 12 and 24 hpi and provide information on how many genes are expressed differently. These genes may be used as target genes for resistant breeding. Provide at least one section on analysis.

Response 1: Ok, based on your feedback, this part has been added to the text. (Please see lines 160-166 of the revised manuscript and figure S2 of supplementary files)

  1. Same kind of analysis for metabolite.These two analyses will provide genes and metabolites information that will be useful for PM resistance knowledge.

Response 2: Ok, based on your feedback, this part has been added to the text. (Please see lines 310-316 of the revised manuscript and figure S11 of supplementary files)

In addition, due to the reclassification of our school's colleges, we have changed “College of Agriculture” to “College of Enology and Horticulture”. (Please see lines 8 and 759 of the revised manuscript)

Round 3

Reviewer 2 Report

Thanks for incorporating all suggestions